# Assessment of Conservation Effectiveness of the Qinghai–Tibet Plateau Nature Reserves from a Human Footprint Perspective with Global Lessons

**Mingjun Jiang [1], Xinfei Zhao [1], Run Wang [2], Le Yin [1] and Baolei Zhang [1,*]**

[1] School of Geography and Environment, Shandong Normal University, Jinan 250014, China; yinl.16b@igsnrr.ac.cn (L.Y.)

[2] Shandong Provincial Territorial Spatial Ecological Restoration Center, Jinan 250014, China

* Correspondence: blzhangsd01@sdnu.edu.cn

**Abstract:** The intensity of human pressure (HP) has an important impact on the biodiversity and ecosystem services of nature reserves (NRs), and the conflict and the coordination between NRs and human activities are now key issues to solve in the construction of NR systems. This study improved and applied a human footprint (HF) model that considers population density, land use, night light, grazing intensity, and road construction as indicators of human activity to evaluate the effectiveness of NRs in the Qinghai–Tibet Plateau in mitigating HP from 2000 to 2020. The results indicated that during this period, the average HP in the national NRs of the plateau increased from 1.47646 to 1.76687, where values were generally high in the east and low in the west. The average value in wetland NRs was the largest and had the smallest growth rate, while that in desert NRs was the smallest and had the largest growth rate. From 2000 to 2020, the average HP in the core areas, buffer areas, and experimental areas of the NRs increased by 0.12969, 0.29909, and 0.44244, respectively. It is a challenge for the Chinese government to strengthen the ability of NRs to mitigate HP on the wetland reserves and experimental zones in the Qinghai–Tibet Plateau region.

**Keywords:** protection effectiveness; nature reserve; human pressure; Qinghai–Tibet Plateau

## 1. Introduction

Nature reserves (NRs) are considered cornerstones of biodiversity conservation, and their number and extent are expanding rapidly worldwide [1,2]. In 2020, more than 200,000 NRs have been established globally, covering 15.4% of the land and 7.5% of the marine environments [3]. While the international community has made important progress toward placing more land under protection, global biodiversity continues to decline, and the effectiveness of NRs has been questioned [4–6]. To address this crucial issue, there is a growing call for empirical evaluations of NRs' effectiveness in achieving their intended conservation goals and to understand the reasons behind their success or failure [7–9]. Climate change and human activities are important factors affecting biodiversity and ecosystem services [10–13]. With the rapid development of global society and continuous population growth, the conflict between protection and economic development has become increasingly prominent, and human activities increased by about 55% in spite of the establishment of more than 20,000 global reserves [14]. The fragmentation of habitat patches caused by human activities such as urbanization, deforestation, and road construction has been recognized as the biggest threat to ecological diversity within NRs [15,16]. Therefore, the assessment of HP is an important means of evaluating their effectiveness.

In recent years, a series of studies has been carried out on the impact of human activities on the ecological environment from the perspectives of biodiversity [17,18], biological habitat [19–21], and ecosystem services and their value [22,23]. With the rapid development of relevant theories, a new method of measuring HP, known as the human footprint (HF)

model, has been recently proposed [24]. This model quantifies the disturbance degree of the ecosystem and then accumulates different human interference factors. It can comprehensively measure multiple HPs on the environment [24,25], as well as the intensity of cumulative disturbance caused by some HP categories to ecosystems, including construction sites, crop and pasture lands, population density, nighttime lights, roads and railways, and navigable waterways [25]. The HF model has been widely used since the beginning of the 21st century [26,27], providing a reference for the study of HP interference [28]. It has also been widely adopted for the protection of NRs in China [29], the Qilian Mountains [30,31], the southeastern coast of Bangladesh [32], and the Yellow River Delta [33], where it was shown to be advantageous to evaluate the effectiveness of reserves. With time, increasing numbers of interference factors have been used as impact factors in the HF model to evaluate HP, including land use [34], road construction [35], grazing intensity [36], energy consumption (such as night light), and population density [37]. However, most studies have only investigated one or two influencing factors, so the assessment results are not perfect. It is urgent to consider all the abovementioned factors to evaluate HP in NRs and fill the gap in the existing data set for this complex parameter.

The Qinghai–Tibet Plateau is the largest plateau in China and an important ecological security barrier in both China and the Asian continent [38]. The plateau is not only a key area for the distribution of alpine ecosystems and endemic animal and plant species but also one of the regions with the richest biodiversity in the world [39]; it also has the highest concentration of threatened terrestrial ecosystems [40]. Since the establishment of the first national NR on the Qinghai–Tibet Plateau in 1963, 150 additional NRs have been created, accounting for 31.63% of the plateau area [41]. In recent years, domestic and foreign research groups have carried out relevant research on species protection, ecological and environmental changes, protection measures, and the effects of NRs in the Qinghai–Tibet Plateau region, where the impact of human activities has become an important part of the evaluation [42,43]. For example, it has been found that the damage to the normal growth of vegetation caused by long-term overgrazing is the main factor leading to grassland degradation in these reserves [44]; road traffic facilities can directly or indirectly modify the habitat of wild animals and plants and can even lead to habitat fragmentation and loss [45]; tourism-related activities will disturb the habitats of animals and plants in the NRs, directly destroy the surface vegetation, and alter the physical and chemical properties of soil [46,47]; and, finally, the development of the mineral extraction industry will have a serious impact on the migration of wild animals in the NRs [48]. In addition, engineering construction and urbanization processes also to varying degrees threaten the NRs of the Qinghai–Tibet Plateau [49]. Although numerous studies have been conducted on HP in the area, as far as we know, no systematic report is available on the current situation in the whole plateau in terms of the effects of constructing NRs, the protection they ensure, and the existing problems. Therefore, by comprehensively considering multiple interference factors examined in previous studies, we can adequately estimate what has been the level of HP in the national NRs of the Qinghai–Tibet Plateau in recent years. Because of the increasing population density and continuous expansion of production and living space in the region, we hypothesized that although HP increases less inside than outside, this parameter would have still slightly increased in the reserves during the period examined. To verify this hypothesis, we comprehensively considered five interference factors to assess HP in the national NRs of the Qinghai–Tibet Plateau.

The main aims of this study were to (1) improve and apply a human footprint (HF) model that considers population density, land use, night light, grazing intensity, and road construction as indicators of human activity to evaluate the effectiveness of NRs in the Qinghai–Tibet Plateau region and (2) compare and analyze the variation of HP values inside and outside the reserves and in each functional area to evaluate the effectiveness of NRs in reducing human impacts from 2000 to 2020. The paper is structured as follows: Section 1 introduces the background of the analysis; Section 2 presents the methodological approach;

Section 3 summarizes the results; and Section 4 discusses those results. Concluding remarks are presented in the final section.

## 2. Materials and Methods

### 2.1. Study Area

The Qinghai–Tibet Plateau (73.43~104.67° E, 25.98~39.82° N), in the southeast of China, is a unique geographical unit with the highest average altitude in China and even in the world, with an average altitude of more than 4000 m. The Qinghai–Tibet Plateau includes Qinghai Province, Tibet Autonomous Region, Sichuan Province, Yunnan Province, Gansu Province, Xinjiang Autonomous Region, and other regions, with a total area of 2,500,000 km², accounting for 26% of China's total land area. The uplift of the Qinghai–Tibet Plateau has blocked the Indian Ocean monsoon from moving northward and the low-level westerly flow in China, forming a unique plateau climate, which combines subtropical, temperate, and other climatic zones with complex hydrothermal conditions. The vegetation types include mainly forests, shrubs, meadows, and deserts, and the alpine meadows are the most widely distributed, accounting for about 60%.

The population density in the Qinghai–Tibet Plateau is relatively low, with less than 2 people/km² in the central and western regions and slightly more than 10 people/km² in the southeast regions. The Qinghai–Tibet Plateau lags in urbanization and industrialization; it is rich in minerals, oil, natural gas, and other energy resources; and its economic development level is lower than the national average. However, its overall development speed is higher than the national average. The construction of NRs in the Qinghai–Tibet Plateau began in 1963, and there were 56 national NRs in 2022. In this research, 32 national NRs completely located in the Qinghai–Tibet Plateau with clear three functional zones were selected as the study objects. The selected NRs were divided into five categories [50], including 4 integrated ecosystems (TIEs), 13 wildlife ecosystems (TWEs), 7 forest ecosystems (TFEs), 2 desert ecosystems (TDEs), and 6 inland wetlands and water ecosystems (TWWEs) (Figure 1).

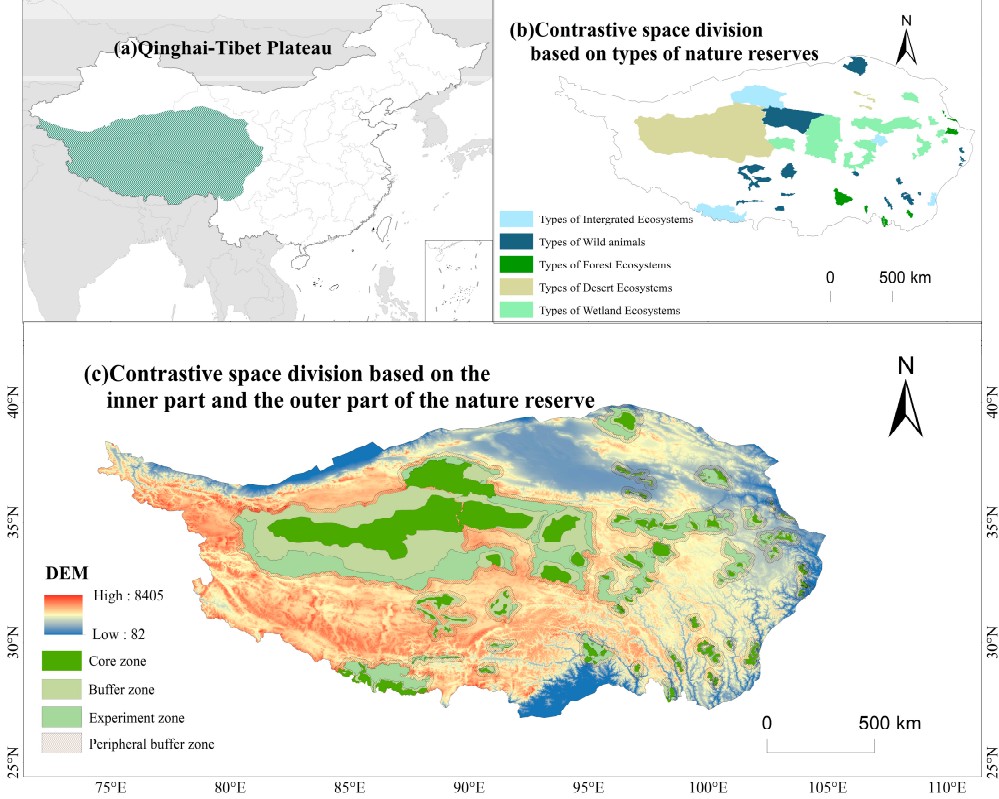

**Figure 1.** Location of the Qinghai–Tibet Plateau and the distribution of NRs and their functional zones.

### 2.2. Data

The data include mainly population density data, land-use data, grazing intensity data, night-light data, road data, etc. The data source is shown in Table 1.

**Table 1.** Data source and time.

| Data Type | Time | Data Sources |
| --- | --- | --- |
| Population density data | 2000–2020 | World Pop data set |
| Land-use data | 2000–2020 | Resource and Environmental Science Data Center of the Chinese Academy of Sciences |
| Grazing intensity data | 2000–2020 | Global Ecosystems and Environment observation Analysis Research Cooperation |
| Night-light data | 2000–2020 | National Oceanic and Atmospheric Administration (NOAA) |
| Road data | 2002–2020 | Open Street Map |

### 2.3. Methods

Based on the local conditions of the Qinghai–Tibet Plateau and the availability of data, this study uses the HF model [25], taking into account five interference factors, namely population density, night light, road construction, land use, and grazing intensity, and first measures the respective HP values of the NRs of the Qinghai–Tibet Plateau. It then compares and analyzes the changes in HP values inside and outside the reserve and in each functional area and evaluates the effectiveness of the national NRs on the Qinghai Tibet Plateau in reducing cumulative impacts on the HP.2.3.1 population density.

The growth of the population density increases the demand for ecosystem services. Therefore, we take population density as an evaluation factor of HP. The population distribution in the Qinghai–Tibet Plateau is uneven, and the population in different regions greatly varies. Logarithmic processing can better represent the variability of population data without changing the nature and correlation of the data. Concerning the global model method [25,51–53], the grid with a population density greater than 1000 people/km$^2$ was assigned 10 points, and the grid with a population density less than or equal to 1000 people/km$^2$ is assigned by the following formula to calculate the population pressure score of the ecosystem:

$$\text{popd}(i,t) = 3.333 \times \log_{10}(\text{popden}(i,t) + 1) \tag{1}$$

where $\text{popd}(i,t)$ is the value of the population pressure intensity of grid $i$ for year $t$ and $\text{popden}(i,t)$ is the population density of grid $i$ for year $t$.

#### 2.3.1. Land-Use Activity

Human land-use activities have great impacts on the ecological environment, so we apply land-cover types to the HF model. 0–10 points were assigned different land-use types on the Qinghai–Tibet Plateau on the basis of relevant research [54,55]. The impact of grazing on the ecosystem in the NRs was considered because there were many pastoral areas distributed in this region. However, by using only satellite-based land-use data, it is difficult to distinguish whether grassland is used for grazing. The grassland was temporarily assigned a value of 0 in this section.

#### 2.3.2. Grazing Intensity

The Qinghai–Tibet Plateau is a traditional grazing area in China and also the most developed area of animal husbandry in China. Grazing has a great impact on the grassland ecosystem. We choose grazing intensity as the interference indicator to describe the grassland ecosystem. The pressure scores caused by grazing intensity are calculated by

normalizing the data to 0–10 through the following formula and on the basis of relevant research on data scale conversion [56]:

$$Norgrad(i,t) = \frac{grazd(i,t) - grazd(i,t)_{min}}{grazd(i,t)_{max} - grazd(i,t)_{min}} * 10 \tag{2}$$

where $Norgrad(i,t)$ is the normalized grazing density of grid $i$ for year $t$, ranging from 0 to 10; $grazd(i,t)$ is the grazing density of grid $i$ for year $t$, which is the rasterized county-level data; and $grazd(i,t)_{max}$ and $grazd(i,t)_{min}$ are the maximum and minimum values of the original data set, respectively.

### 2.3.3. Nighttime Light Activities

Nighttime lights indicate how much electrical energy is consumed. Its spatial resolution is 1 km, and the years are from 2000 to 2020. The nighttime light images were calibrated by using the pseudoinvariant pixel method [57] and were grouped by using the classification interval determined by the natural discontinuities method, and the disturbance scores to the ecosystem were assigned a value from 0 to 10.

### 2.3.4. Distance from Road

The impact of roads on ecosystems accounts for at least 15~20% of the global land [58]. Roads and railways have great impacts on the surrounding ecosystems [59–61], and the interference distance even reaches 5 km [61]. Owing to the lack of road data in 2000, this paper selects road data in 2002 to represent roads in 2000. According to the global human footprint data set [25,62], we classify roads as a category of HP and assign different scores, on a 0–10 scale, to different levels of road buffers (Table 2).

**Table 2.** Human influence scores of different road levels on the ecosystem.

| Road Categories | Buffer Distance | | |
|---|---|---|---|
| | 0–1 km | 1–2 km | 2–5 km |
| Freeways | 10 | 6 | 3 |
| National roads | 8 | 4 | 2 |
| Provincial roads | 4 | 2 | 1 |
| County roads | 2 | 1 | 0 |
| Railways | 8 | 4 | 1 |

### 2.3.5. HP Calculation

Five interference factors of human pressures, i.e., population density, grazing intensity, land-use intensity, road construction, and nighttime lights were used to map HP according to the characteristics of the study area and data availability [25]. These interference factors were quantified by their disturbance degree to the ecosystem and then cumulatively summed. The spatial resolution of the HF map was determined as 1 km by using all available data sets of human pressures. Their disturbance to the ecosystem was quantified in the range of (0, 10) (0 is the minimum disturbance and 10 is the maximum; see Sections 2.3.1–2.3.5 for details) and accumulated in equal weight to measure the HP values. The equation was as below [51,52]:

$$HP(i,t) = \text{popd}(i,t) + \text{landuse}(i,t) + \text{graz}(i,t) + \text{road}(i,t) + \text{nightlight}(i,t) \tag{3}$$

where $HP(i,t)$ is the HP value of grid $i$ for year $t$ and $\text{popd}(i,t)$, $\text{landuse}(i,t)$, $\text{graz}(i,t)$, and $\text{nightlight}(i,t)$ are the disturbance intensities of population density, land use, roads construction, and energy consumption to the ecosystem in grid $i$ for year $t$, respectively.

*2.4. Changes in the Values of HP*

The changes of the *HP* values were compared within the NRs of the Qinghai–Tibet Plateau for the period 2000–2020 by using the following equation:

$$\Delta HP(i, \Delta t) = HP(i, t_m) - HP(i, t_n) \tag{4}$$

where $\Delta HP(i, \Delta t)$ refers to the change in the HP value of grid *i* from $t_m$ year to $t_n$ year.

In addition, the types of nature reserves and different functional areas have different management requirements for human activities. Therefore, we further analyzed *HP* changes in different types of national NRs and three functional areas of the reserves on the Qinghai–Tibet Plateau from 2000 to 2020.

## 3. Results

*3.1. Spatiotemporal Changes in the HP in the NRs of the Qinghai–Tibet Plateau for 2000–2020*

From 2000 to 2020, HP on the national NRs of the Qinghai–Tibet Plateau has been high in the east and low in the west. Specifically, the NRs with low HP were in the western region thanks to its high altitude and harsh environments. HP in the NRs was generally low, with an average of 1.6434, reaching the lowest and highest values in 2005 and the highest value in 2015, respectively, and the variation exhibited an overall upward trend.

The average HP value in the NRs increased from 1.4765 in 2000 to 1.7669 in 2020, indicating an overall increasing trend throughout the period examined. The value decreased from 1.4765 in 2000 to 1.3326 in 2005, and next, the average increased to a maximum of 1.9326 in 2015. From 2015 to 2020, it slightly declined, reaching 1.7669 in 2020. In terms of spatial variation, although higher values were reported in the east, in general, the spatial pattern was relatively stable from 2000 to 2020 (Figure 2).

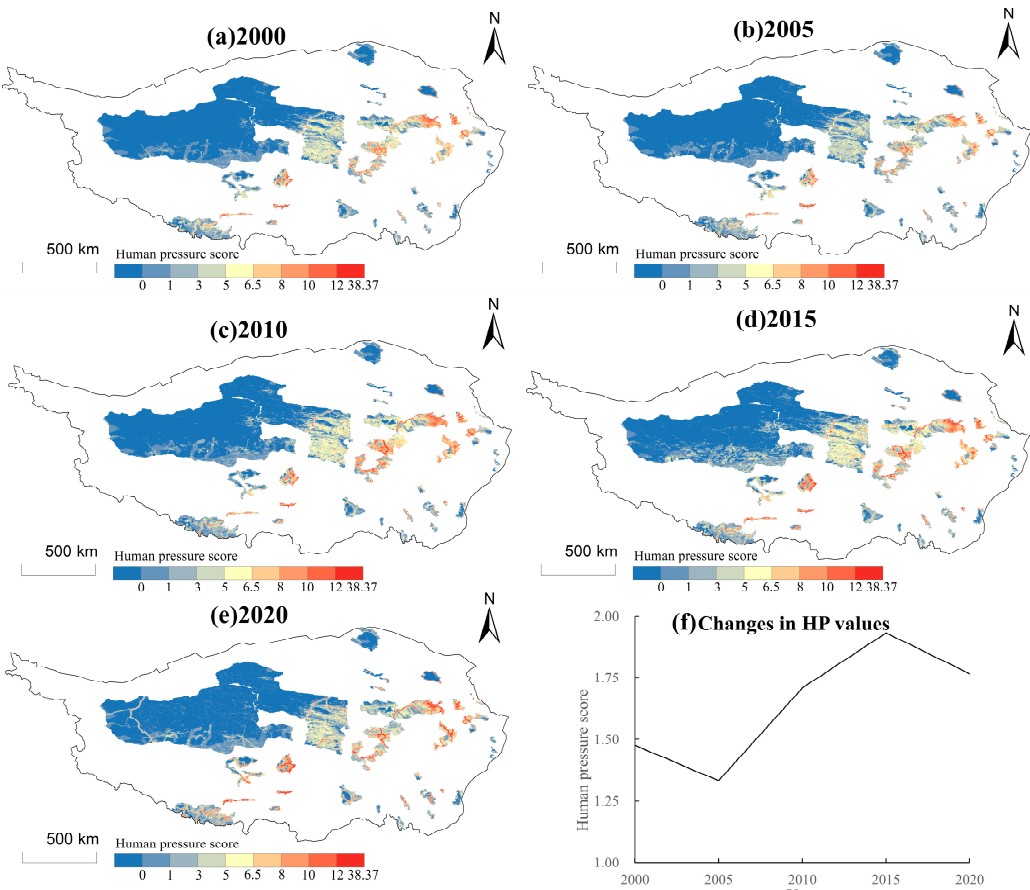

**Figure 2.** Spatiotemporal changes in HP in the NRs of the Qinghai–Tibet Plateau for 2000–2020.

In terms of the interference value of the five HP factors considered in this study, population density, land-use intensity, night light, and road construction showed an overall upward trend from 2000 to 2020. In particular, in this period, the pressure value of roads and the pressure value of population density increased by 0.2523 and by 0.10272 from 2000 to 2020, respectively. However, grazing intensity decreased year by year after increasing. This parameter showed an upward trend from 2005 to 2015, reaching the maximum value of 1.9326 in 2015 and decreasing to 0.8335 in 2020 (Table 3).

**Table 3.** Disturbance values of five HPs in the NRs of the Qinghai–Tibet Plateau for 2000–2020.

| Human Pressures | 2000 | 2005 | 2010 | 2015 | 2020 |
|---|---|---|---|---|---|
| Roads | 0.1213 | 0.1343 | 0.1397 | 0.1556 | 0.3738 |
| Nighttime lights | 0.0001 | 0.0002 | 0.0005 | 0.0006 | 0.0017 |
| Population density | 0.4102 | 0.4126 | 0.4178 | 0.4864 | 0.5129 |
| Land-use activity | 0.0377 | 0.0380 | 0.0380 | 0.0383 | 0.0655 |
| Grazing intensity | 0.9254 | 0.7731 | 1.1212 | 1.2757 | 0.8335 |
| HP value | 1.4765 | 1.3326 | 1.7085 | 1.9326 | 1.7669 |

The analysis of the spatial variation of the five HP interference types, from 2000 to 2020, showed that the overall changes in nighttime lighting and land use were relatively small, and the grazing intensity in most areas of the eastern region showed a downward trend; in contrast, the population density and road construction generally showed a significant upward trend in the same region. Overall, HP increased in the NRs in the eastern and southern parts and decreased in those in the central part of the Qinghai–Tibet Plateau (Figure 3).

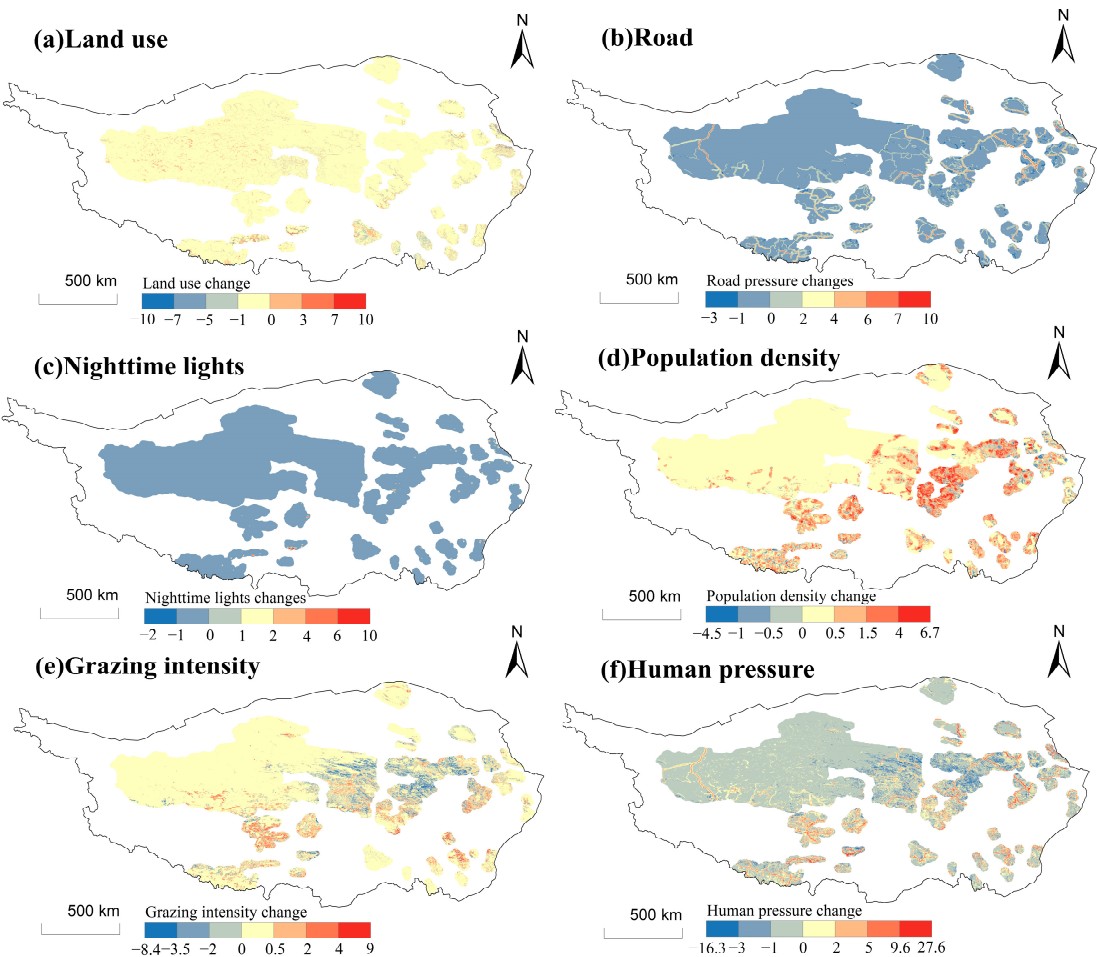

**Figure 3.** Spatiotemporal changes in five HPs in the NRs of the Qinghai–Tibet Plateau for 2000–2020.

### 3.2. HP Changes inside and outside the NRs of the Qinghai–Tibet Plateau

In the late 1990s and early 21st century, large national NRs such as Sanjiangyuan, Lalu Wetland, and Hoh Xil were established in the Qinghai–Tibet Plateau. From 2000 to 2005, the pressure of population growth in the internal and external buffer zones of the NRs showed a decreasing trend, which was greater outside than inside the national NRs, and was observed mainly in the central and eastern reserves of the plateau, especially in the Sanjiangyuan National NR (Figure 4). From 2005 to 2015, HP values in the internal and external buffer zones of the NRs increased, and in the latter, the values were considerably higher. From 2015 to 2020, the trend in both zones also increased because of road construction, and HP values increased less in the internal NR zones. From the aspect of spatial changes, HP mainly increased in the southern Qinghai–Tibet Plateau from 2000 to 2020.

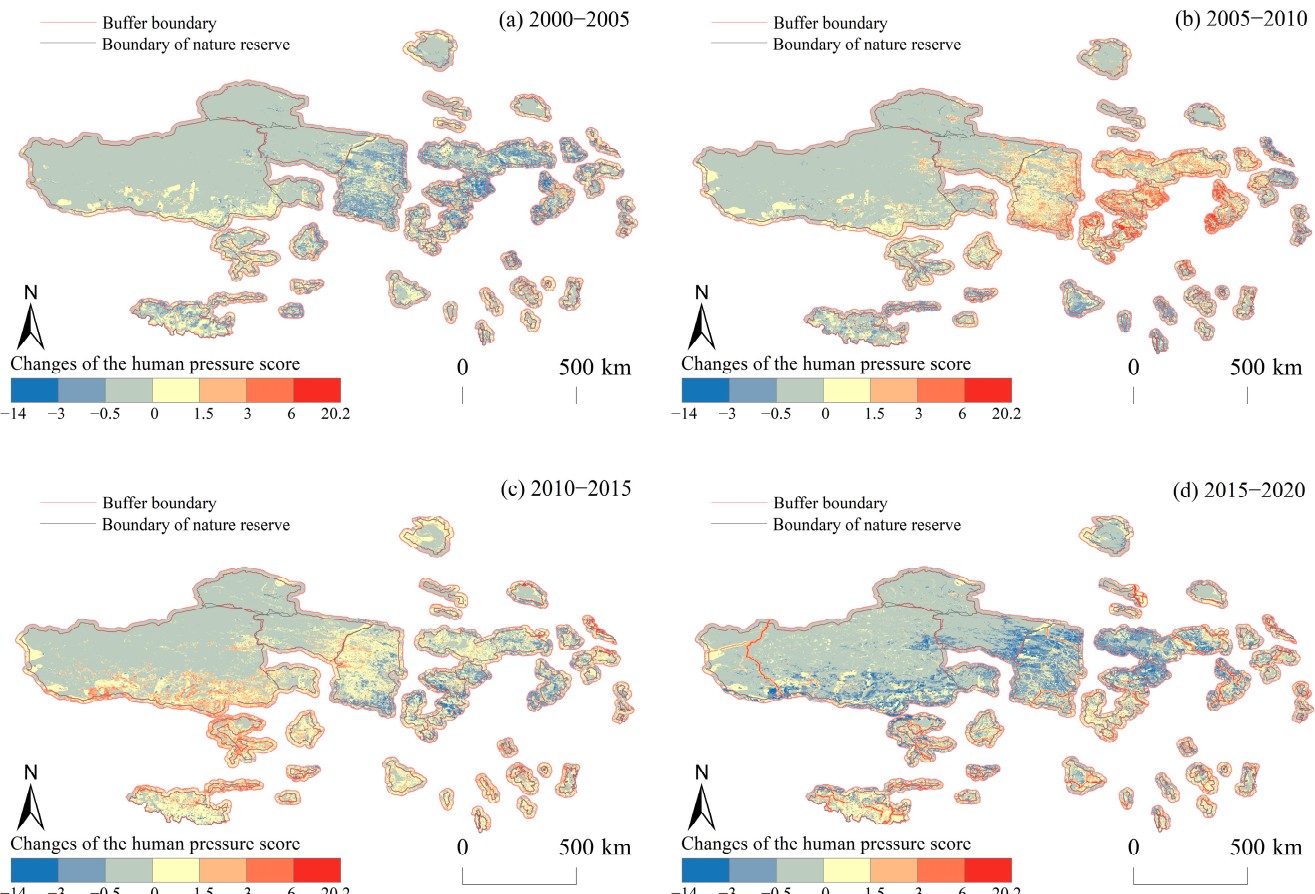

**Figure 4.** Spatiotemporal changes in HP inside and outside the NRs of the Qinghai–Tibet Plateau for 2000–2020.

The national NRs of the Qinghai–Tibet Plateau are partly positive in reducing the interference in the five HP types. From 2000 to 2020, the HP values for road construction were always lower inside the NRs than in the external buffer zones (Figure 5). From 2000 to 2015, these values inside the NRs and inside the external buffer zones increased by 0.0343 and 0.2024, respectively. However, from 2015 to 2020, the same parameter inside and outside the NRs increased by 0.2181 and 0.2854, respectively. The values inside and outside the NRs showed almost the same large increase, indicating that the regulatory capacity of protection measures to mitigate road construction decreased during these five years. The increase in night-light pressure was always lower in the NRs than in the external buffer zones from 2000 to 2020, and the increase rate was small (Figure 5). This indicated that the energy conservation measures adopted in the NRs effectively regulated the disturbance associated with of energy consumption in both the internal and external buffer zones.

From 2005 to 2010, population density without the NRs showed a decreasing trend, while the internal population density showed a growing trend (Figure 5), indicating that the measures aimed at controlling population growth did not work during the 2005–2010 period, but the effect of controlling population growth outside the NRs was good. During the 2000–2005 and 2010–2020 periods, population density always increased less in the NRs than in the external buffer zones, indicating that the plan to control population growth in the NRs was effective during these periods.

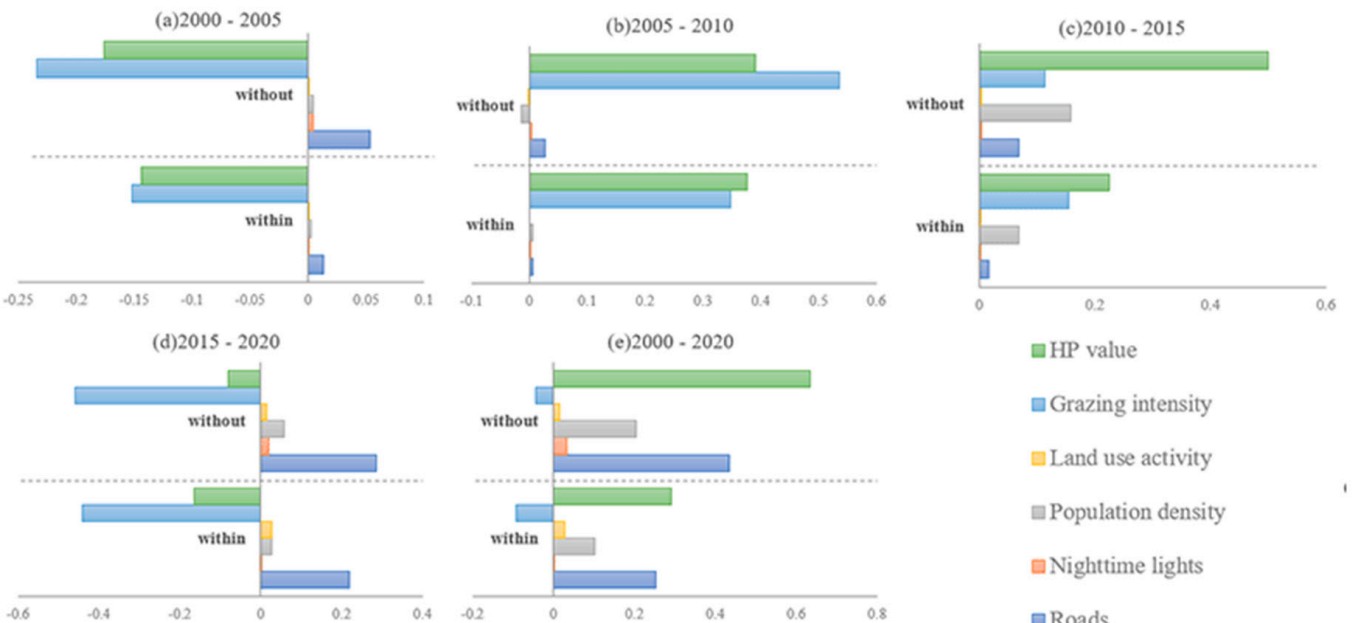

**Figure 5.** Changes in HP values inside and outside the NRs of the Qinghai–Tibet Plateau for 2000–2020.

From 2005 to 2010, land-use intensity in the internal and external buffer zones of the NRs showed a downward trend, and the decline in the latter was even greater (Figure 5), indicating that the protection measures were successful both inside and outside the NRs during this period. However, except for these 5 years, the land-use intensity showed an increasing trend in the rest of the research period, and from 2015 to 2020, the increase was greater in the NRs than in the external buffer zones. This shows that the proximity effect of the NRs is good but that protection measures still need to be strengthened.

Grazing intensity fluctuated and decreased from 2000 to 2020 (Figure 5). From 2005 to 2015, it significantly increased inside and outside the NRs. Moreover, thanks to the implementation of grazing prohibition measures, the situation improved from 2015 to 2020, where this parameter decreased to levels below those reported in 2000.

Overall, these results showed that the implementation of measures to reduce HP in the NRs of the Qinghai–Tibet Plateau has been partially effective. From 2000 to 2020, although all the HP factors except for grazing intensity showed an increasing trend inside the NRs, the increase was far lower than that in the external buffer zones. In 2020, the HP values in the internal and external buffer zones were 1.7669 and 4.2988, respectively. This huge difference indicates that the implementation of protective measures in the plateau's NRs was effective.

### 3.3. Temporal Variation in HP in Different Types of the NRs

The HP on different NR types in the Qinghai–Tibet Plateau increased from 2000 to 2020 (Figure 6), where the values in desert reserves exhibited the highest growth rate, 169.04%, and the lowest HP value, only 0.3316. HP in wetland NRs had a value of 4.0317 and showed the lowest growth rate, 5.23%. In comprehensive ecological reserves, wildlife

reserves and forest reserves the HP values were 1.4898, 2.2645, and 3.0886, respectively, from low to high, with growth rates of 22.79%, 33.78%, and 38.16%, respectively. HP values in all the NR types, except for the desert type, showed a downward trend from 2000 to 2005 and increased during during the 2005–2015 period.

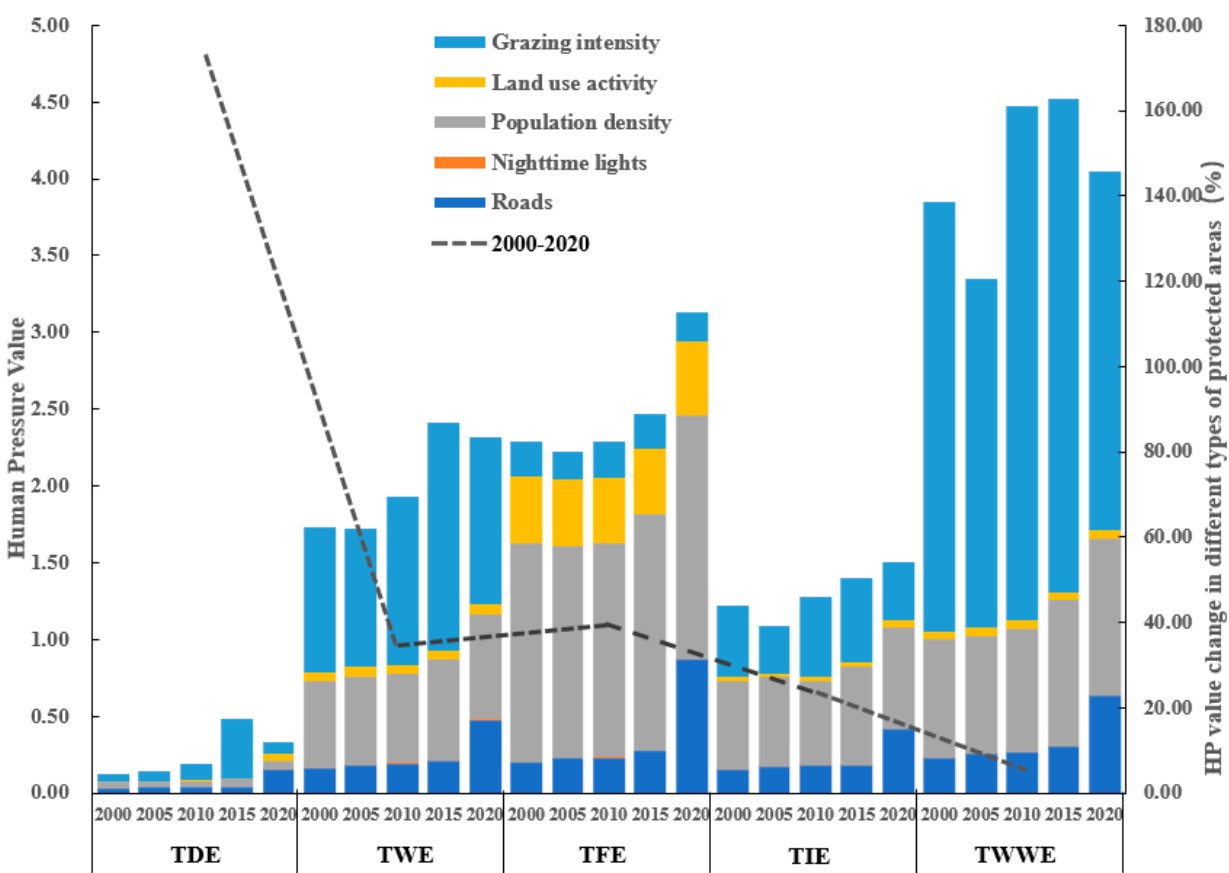

**Figure 6.** Temporal variation in HP in different types of the NRs for 2000–2020.

The increase in population density in all five types of NR was small, and in the comprehensive ecological reserves and forest reserves, this parameter even showed slight downward trends. The desert, wild animal, and wetland reserves significantly increased from 2005 to 2015 and then showed a downward trend from 2015 to 2020. The HP associated with land use was higher in forests than in other reserve types, and this was due mainly to the cultivation of farmland in the former. From 2000 to 2015, the interference from road construction in the five reserve types steadily increased, with a significant spike from 2015 to 2020, especially in forest reserves, where the value increased by 216.27%. Night-light interference accounted for a minimal proportion of HP in the different NRs and could therefore be ignored. The temporal variation in HP in different NR types showed that the current protection measures have effectively controlled population growth and the use of light at night in these reserves; however, road construction, grazing, and land-use activities, as well as production and daily living activities, still increased during the period examined.

*3.4. Changes in HP in Various Functional Areas of the NRs*

In each same period, the HP values in the three functional areas decreased, moving from the experimental area to the buffer area and core areas, in that order. These values showed a downward trend from 2000 to 2005, and the largest drop was detected in the experimental area (0.2321), followed by the core area and buffer area, in that order. From 2005 to 2015, the HP values in the three functional areas all increased, with the lowest increase detected in the core area, followed by the buffer area and experimental area

(Figure 7). From 2015 to 2020, the values declined in three functional areas, and the decrease was largest in the experimental area, followed by buffer area and core area, due mainly to the reduction in grazing intensity.

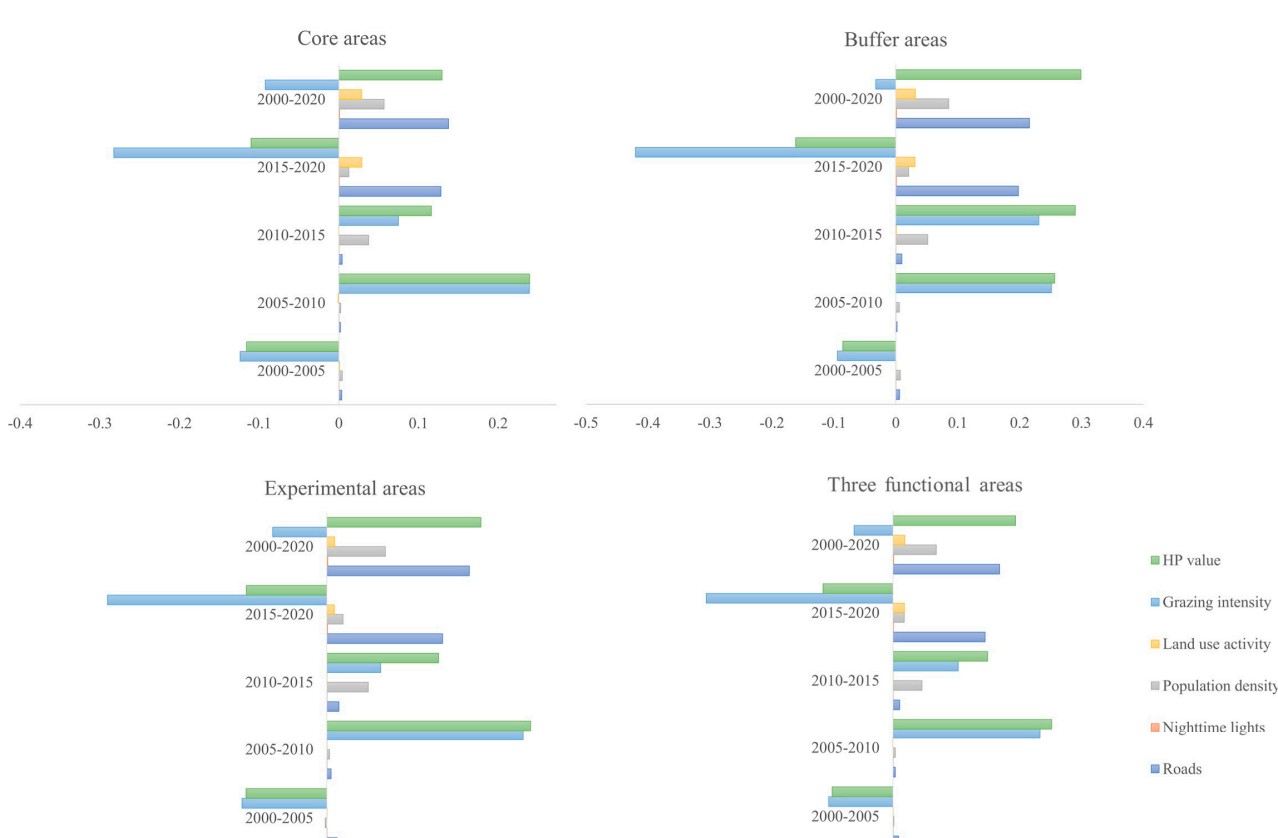

**Figure 7.** Changes in HP in various functional areas of the NRs for 2000–2020.

In the core zones of the NRs, grazing intensity significantly increased from 2000 to 2015 and then significantly decreased by 0.2825 from 2015 to 2020, which was still much lower than the decrease observed in 2000 (Figure 7). Land-use intensity from 2005 to 2015 was reduced to a better degree compared with that in the 2015–2020 period; the HP values for population density, road construction, and night light in 2020 were higher than those in 2000.

In the buffer zones of the NRs, grazing intensity was reduced by 0.2825 from 2015 to 2020 (Figure 7). The HP values associated with the intensity of land use, road construction, population density, and night light were all on the rise from 2000 to 2020, the most significant being road construction value, which increased by 0.2157.

In the experimental zones of the NRs, population density and grazing intensity were effectively alleviated from 2000 to 2005, and the HP values associated with each of them decreased by 0.005 and 0.24 37, respectively (Figure 7). The land-use intensity, road construction, population density, and night-light values increased by 0.4091, 0.0029, 0.1681, and 0.0227, respectively from 2000 to 2020, which indicated that the mitigation effect of protection measures on these four HP values was poor, especially for road construction from 2015 to 2020.

## 4. Discussion

### 4.1. Analyses of Differences in Conservation Effectiveness

The results of this study showed that the overall HP value on the NRs of the Qinghai–Tibet Plateau decreased from 2000 to 2005 and from 2015 to 2020, which indicates obvious

positive effects in mitigating human activities in these two periods. Similarly, the protection measures adopted in the reserves were also partially effective against the five types of human interference factors. HP on the NRs of the Qinghai–Tibet Plateau was high in the east and low in the west, results that are in line with recent studies, specifically with the global HF data reported in Venter [25].

Grazing intensity was significantly reduced during the 2000–2020 period because China implemented large-scale grassland restoration and ecological protection projects at the beginning of this century. Thanks to the launching of these projects, the number of livestock in and around the NRs has been reduced and grazing intensity alleviated. From 2005 to 2010, land use was effectively regulated, but it was not reduced to the original level. The main reason was that the economy of the Qinghai–Tibet Plateau developed rapidly, and the secondary and tertiary industries increased year by year. Especially after 2010, agricultural production activities decreased, driving the reduction of cultivated land, which has been converted into grassland, urban land, and forest land [63]. The reason was that the engineering buildings left behind by the original human production were still in the NRs [64]

China's implementation of the large-scale policy of returning farmland to forests at the end of the 20th century, and the prohibition of grazing as well as the enclosure of degraded grasslands at the beginning of the 21th century, represented incentives and subsidy policies for the protection of grasslands [65,66]. Road construction was greatly developed in the NRs from 2015 to 2020, due mainly to the construction of the G216 National Highway, which started in 2016 and spanned the Qiangtang NR. The building of both trunk roads and surrounding branch roads was necessary, and this led to an expansion of the road network in the reserves [62].

In terms of different functional areas, the growth of the HP index from 2000 to 2020 was considerably lower in the core zones than in the buffer zones and experimental areas. In the core zones, HP associated with land use showed a downward trend from 2005 to 2015, indicating that protection measures contributed to the growth of vegetation [67] and increased the coverage of the ecosystem [68]. Not only has grazing intensity decreased in the buffer and experimental zones, but the increase in the HP values associated with energy consumption, population density, land use, and grazing was considerably lower in these zones than in the external buffer zones. This was due mainly to the government's actions, which reduced the pressure of production and daily activities inside the NR [69].

This study also showed that the overall HP was considerably higher in wetland NRs than in the other four NR types examined. Specifically, grazing intensity and population density accounted for a large proportion of HP because wetland nature reserves are distributed mainly in the eastern part of the Qinghai–Tibet Plateau. Compared with the other regions of the Qinghai–Tibet Plateau, the eastern region is relatively flat with abundant rainfall [70]. The climate was therefore suitable for pasture growth [71]. In particular, population density in the Sanjiangyuan National NR increased year by year [56]. HP in desert NRs was the lowest, which was due mainly to the harsh climate and environment, which was not suitable for grazing and human living. Land use and population density in forest NRs accounted for a large proportion of HP, mainly because of the large area of cultivated land in these reserves and the high levels of production and living activities [72].

To achieve conservation objectives, the government must strengthen the management of NRs. China's implementation of the large-scale policy of returning farmland to forests at the end of the 20th century, as well as the prohibition against grazing and the enclosing of degraded grasslands in 2009, represented incentives and subsidy policies for the protection of grasslands. Although grazing intensity in the NRs of the Qinghai–Tibet Plateau was gradually reduced, the cultivated land area and road construction still increased [73]. To further achieve the protection goals, since the beginning of the 21st century, local governments have vigorously pursued ecological protection policies, such as returning farmland to pasture, enclosure protection, and ecological migration, so as to improve the vegetation coverage of the NRs and their surrounding areas [74]. In response to the national policy of

ecological civilization construction and development, in 2009 Tibet's government issued the Plan for the Protection and Construction of Tibet's Ecological Security Barrier (2008–2030), which divided the area into prohibition-restricted and conditional development zones. Researchers should also pay attention to ecological protection and water conservation.

The significance of the present study in terms of policies is that it recommends the implementation of more measures to manage NRs in China. Specifically, it is necessary to regularly assess the intensity of HP in and around the reserves, especially in the western region, where human activities continue to increase thanks to the urgent need for economic development. However, to protect biodiversity and natural resources without considering the quality of life of local residents is also inappropriate. Therefore, it is necessary to reasonably set the boundary of each reserve, establish the reserves in areas with high ecological service values, and reasonably develop the economy in the areas with low ecological service values [2].

*4.2. Limitations and Future Works*

While a data set for HP in the Qinghai–Tibet Plateau National NRs was developed in this study, a number of limitations were noted. First of all, the road construction data for the plateau region were missing for year 2000 and were therefore replaced with the data from 2002. Moreover, the road construction data after 2010 were obtained from OSM (Open Street Map), and there was no clear basis for the classification of roads below the provincial level. Therefore, more-accurate data were needed to correct possible errors. In addition, different measures implemented in different functional areas would have varying efficiencies. However, the existing information on ecological measures was related mainly to the overall evaluation of indicators, and there were only a few of these for each functional area. It is recommended that future studies obtain more data on ecological measures for different functional areas. Moreover, because HP is evaluated on the basis of different criteria in each NR type, this study did not consider the protection requirements of different NRs in detail, which needed further attention. Finally, more HP types could be included in the data set. In recent years, production activities, such as ore mining, tourism, livestock breeding, and other construction and development activities in the Qinghai–Tibet Plateau, especially in the Sanjiangyuan National NR, have grown rapidly [75]. Because of the difficulty in quantifying them, these activities have not been considered in this study; however, if they occur outside of protected areas, they will also pollute water sources in wetlands [76]. In the future, the impact that these human activities that take place outside the NRs have on the reserves should be comprehensively considered to obtain more-accurate and more-reliable estimates of the effectiveness of NRs.

This study showed that the correlation between population density and HP factors (such as land use) was high, and the absolute HF value obtained by direct equal weight space accumulation may also be high [77], especially in urban areas with high population density. In the future, spatial accumulation using the fuzzy algebraic sum method [78] may be used as an alternative method to obtain a more-accurate spatial mapping of the intensity of human activities.

Previous studies have shown that the application of the HF model to evaluate the effectiveness of NRs in China is highly feasible [25], and the results are accurate and reliable. Although the five HP categories used in the present study presented the problem of collinearity, the cumulative HP value did not change the overall trend for each of them. The model, which is easy to run, has already been used to evaluate the effects of some protected areas [52–54], and it is recommended to continue to adopt it to evaluate the effectiveness of all the protected areas in the vast territory of China and other parts of the world.

**5. Conclusions**

On the basis of the interference factors associated with human activities in the Qinghai–Tibet Plateau, this study evaluated the HP in the national NRs of the region from 2000 to

2020, specifically selecting road construction, night lighting, grazing intensity, land use, and population density as indicators. Variations in HP values were compared inside and outside the NRs and in each functional area to evaluate the effectiveness of the reserves in reducing human impacts during the period examined. The results showed that the average HP value in the NRs of the Qinghai–Tibet Plateau increased from 1.47646 in 2000 to 1.76687 in 2020. The increase in the road construction and population density values significantly contributed to the overall increase in HP, while the contribution of grazing intensity had a negative value. As for the NR types, the average HP in wetland NRs was the highest, and the growth rate was the lowest; population density and grazing intensity accounted for a large proportion of HP, which was the main reason for the high level of HP reported in wetlands. From 2000 to 2020, the increase in population density in the eastern and southern parts of the Qinghai–Tibet Plateau was the main reason for the rapid increase in HP in these regions. In the same period, the average increase in HP in the NRs' core areas, buffer areas, and experimental areas was 0.12969, 0.29909, and 0.44244, respectively. The increase in road construction pressure in the buffer zones was 0.21574, leading to an overall increase in HP in the NRs of the plateau. More efforts should be made to control human activities in wetland NRs and in the experimental zones of all NRs in the Qinghai–Tibet Plateau.

**Author Contributions:** Conceptualization, M.J. and X.Z.; methodology, M.J.; formal analysis, M.J.; investigation, L.Y.; resources, R.W. and L.Y.; data curation, M.J.; writing—original draft preparation, M.J. and X.Z.; writing—review and editing, M.J. and B.Z.; visualization, B.Z.; supervision, L.Y.; project administration, L.Y.; funding acquisition, B.Z. All authors have read and agreed to the published version of the manuscript.

**Funding:** This research was supported by the National Social Science Foundation of China [18BJY086] and the Natural Science Foundation of Shandong Province, China [ZR2021QD127, ZR2021ME203]. The authors gratefully acknowledge the anonymous reviewers and the members of the editorial team who helped to improve this paper through their thorough reviews.

**Data Availability Statement:** Not applicable.

**Conflicts of Interest:** The authors declare no conflict of interest in the publication of this paper.

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
