# Peer review of "Assessment of Conservation Effectiveness of the Qinghai–Tibet Plateau Nature Reserves from a Human Footprint Perspective with Global Lessons"

_land, doi:10.3390/land12040869_

Round 1

Reviewer 1 Report

Manuscript Title: Assessment of Conservation Effectiveness of Tibetan Plateau Nature Reserves from Human Footprint Perspective

Manuscript ID: land-2305574

This paper evaluates the effectiveness of nature reserves (NRs) in mitigating human pressure in the Qinghai-Tibet Plateau from 2000 to 2020. The study uses a human footprint (HF) model that considers population density, land use, night light, grazing intensity, and road construction as indicators for human activities. The results show that the average value of human pressure in the national NRs of the Qinghai-Tibet Plateau increased from 1.47646 to 1.76687 from 2000 to 2020. The study also found that the average value of human pressure in wetland ecological NRs was the largest with the smallest growth rate, while the average value of human pressure in desert ecological NRs was the smallest with the largest growth rate. The study concludes that it is a challenge for the government to strengthen the ability of the National NRs on the Qinghai-Tibet Plateau in mitigating the human pressure on the wetland ecological reserves and experimental zones.

Overall manuscript presents a crucial research topic, objectives are defined and addressed well with support of good findings. In my opinion manuscript need fewer changes that should be performed carefully by authors.

In general heading format is not same, some headings are capitalized while some are lowercase. There are some grammar mistakes in paper, so it is recommended that read and correct them judiciously. Below are some comments which can be helpful in improving your paper sciebtifically.

l  Section 1: Introduction

1.       Read and revise TYPOS there are some unnecessary capitalize word and spaces.

l  Section 2: Materials and Methods

1.       Line 123-124: NRs full name and acronym are not matching, then also mention same things in figure 1. More detail is required for figure 1, add (a), (b), (c) on figure top. Also add each sub-figure legends in it (if necessary).

2.       From method part extract data and write a separate section for data you used. In addition, it is required to add temporal picture of different Human activities so that reader can physically observe the temporal changes due to 5 human activities (pop. Density, roads, nightlight etc).

3.       Line 153-155: Move these lines in section 2.2.3.

4.       Line 155: “The grassland was temporarily assigned a value of 0 in this section.”, why a value of 0 for grassland is it based on reference if not elaborate it.

5.       Line 175: OSM stand for?

6.       Line 182: What is total score (i.e., for 0-10km railway score is 8/x, what is x?)

7.       Line 191: From equation 3, landuse(?,?)+graz(?,?)+road(?,?)+nightlight(?,?) also provide equations how you calculate value for these terms.

l  Section 3: Result

1.      Figure 2 (f) add y-axis title, and put the legends below all sub-figures horizontally. Figure title can be improved.

2.      Figure 5 correspond x-axis title with changes in line 123-124.

3.      Figure 6 axis title are so congested, picture quality is low. 

l  Section 3: Discussion

1.       More reference can be added to enhance validity of your results

l  Section 5: Conclusion

1.       Line 474: 2010–2020?

2.       The conclusion section is mainly describing changes in value. It should include main finding and conclude what are main factors behind high and low changes in different areas? Which factor is more significant in spatial/temporal changes.

Author Response

Dear Reviewers,

        Thank you for your comments on our manuscript entitled “Assessment of Conservation Effectiveness of the Tibetan Plateau Nature Reserves from a Human Footprint Perspective with global lessons” (Manuscript ID: land-2305574). After carefully studying the comments and your advice, we have made corresponding changes to the best of our abilities. We have marked the revised portions in Red in the manuscript. We also polished the language by resubmitting the revised manuscript to International Science Editing (http://www.internationalscienceediting.cn/) for refinement. We hope that the revised version of the manuscript is now acceptable for publication in Land. Please see to the attachment below for the main comments and our specific response.

Reviewer 2 Report

Assessment of Conservation Effectiveness of the Tibetan Plateau Nature Reserves from a Human Footprint Perspective with global lessons.

This is a useful and well-informed paper. The title might be improved, as suggested in red.

 The paper is about the cornerstone of biodiversity namely the protection given to nature reserves. There is a short abstract, followed by an introduction setting out the methodology of the research.  The human footprint model is used to evaluate   multiple human pressures on the environment.  The Qinghai-Tibet Plateau is the largest plateau in China and is an important  barrier in China and even  in Asia. It is a matter of great significance to the world as well as to the region.

The study has several aims. The first is to refine and improve the application of the  human footprint model, especially when it considers population density, land use hight light as well as  transport and  grazing. The second aim is to compare and contrast  the  changes within the  nature reserve with those that are experienced outside. This seems to be a sensible way to conduct the research an one that is likely to give a more accurate analysis of the impact of the human footprint model as a means of tracking and  assessing any impact on nature conservation. The paper  should be read as providing a case study but beyond that some valuable lessons on the application of the human footprint model. The designation of National Reserves dates from 1963 and since then by 2022 thee are 56 National Reserves. There are a wide range of categories, wildlife nature, forest ecosystems, desert ecosystems,  and wetland reserves.  This is well illustrated  in page 4 of the paper.

Page 4 contains a good explanation of the methodology used for data collection and analysis. Page 5  is critical as it sets out the Human Pressure calculation and how this is arrived at.

Page 6 contains the main results of the study. This is well presented with excellent diagrams and tables that present the information in a clear an logical way. This continues to page 11. There is a useful discussion of the main conservation effectiveness of Nature Conservation. It would be useful to have effectiveness defined  more clearly with a paragraph setting this out.

On page 12 there is discussion of various approaches across different  conservation areas. There is an important analysis of the government’s intervention through  better management schemes and this might be defined in more detail. There is  a useful analysis of limitations on the research and future  works on page 13.

The concluding section on page 13 contains the main conclusions in the paper. It makes depressing reading. The final sentence argues that more efforts are needed to  control human activities.  This is an understatement. It contains  a strong implication that better monitoring is required and that human stresses on the environment are likely to increase.

Author Response

(The authors gave the same response as above.)

Reviewer 3 Report

This manuscript applies human footprint to evaluate the effectiveness of the nature reserves in Qinghai-Tibet Plateau in mitigating human pressure from 2000 to 2020. The results promote our understanding of the National nature reserves on the Qinghai-Tibet Plateau. It's clear from the authors' description that this study is a pseudo-replication, while the results are interesting and worth publishing. Overall, the manuscript should be revised before publication publishing. 

1 Introduction:

(1) Check [19] [20,21] (line 45).

(2) “The Qinghai-Tibet Plateau, known as the "roof of the world", "the third pole" and "the water tower in Asia", is the largest plateau in China and an important ecological security barrier in China and even in Asia” (line 64-66). The Reader who interested in this study was already known that “the Qinghai-Tibet Plateau is the roof of the world, the third pole and the water tower in Asia, is the largest plateau in China”, please deal with lines 26-27.

(3) The use of the term " Qinghai-Tibet Plateau " needs to be checked and standardized because it appears in the text under several distinct names, including Tibetan Plateau.

(4) Hypothesis is missing. Please give your hypothesis.

(5) The description of the introduction is not ideal.

2. Materials and Methods:

(1) “Qinghai-Tibet” (line 106). I guess this refers to Qinghai Province and Tibet Autonomous Region.

3. Discussion

(1) This manuscript used population density, land use, night light, grazing intensity, and road construction to consider human activities. But I did not see the effects of those indicator clearly.

(2) “The finding that the average human pressure of national NRs in the Qinghai-Tibet Plateau was relatively low, which was consistent with the results of Du et al. [65]”. Previous work has shown similar results to this study. What is the significance of this study?

(3) “At the end of the 20th century, China's implementation of the large-scale policy of returning farmland to forests and the implementation of grazing prohibition and enclosure for degraded grasslands in 2009 were the current incentive and subsidy policies for grassland ecological protection”

4. “6. Patents” (line 488). Check it.

5. References

(1) Please check carefully. eg. “Environmental science and pollution research international” (line 508-509), “Acta Geogr Sin” (line 587)….

(2) The reference 44 and 45 are the same one.

6. English presentation:

(1) Language needs editing.

Author Response

(The authors gave the same response as above.)

Round 2

Reviewer 1 Report

Accept

Author Response

Dear Reviewers,

        Thank you for your comments on our manuscript entitled “Assessment of Conservation Effectiveness of the Qinghai-Tibet Plateau Nature Reserves from a Human Footprint Perspective with global lessons” (Manuscript ID: land-2305574). These comments are fair, encouraging, and very constructive for improving our manuscript and future research. We have made corresponding changes to the best of our abilities.  We hope that the revised version of the manuscript is now acceptable for publication in Land. Thank you again for your assistance with our manuscript.

Reviewer 3 Report

1. Authors used “Tibetan Plateau” in the title while used “Qinghai-Tibet Plateau” in other parts.

2. “The average HP on the NRs was relatively low, which was consistent with the results of Du et al.[64]” Please delete this information.

3. References

I am very disappointed that authors did not check the references carefully and there were still many uncertainties. Such as Xiangzheng, D.; Haiming, Y. (line 548), Wang, J.z.; (line 556), Zhang, C.y.; (line 558)…..

Author Response

Dear Reviewers,

Thank you for your comments on our manuscript entitled “Assessment of Conservation Effectiveness of the Qinghai-Tibet Plateau Nature Reserves from a Human Footprint Perspective with global lessons” (Manuscript ID: land-2305574). These comments are fair, encouraging, and very constructive for improving our manuscript and future research. After carefully studying the comments and your advice, we have made corresponding changes to the best of our abilities. We have marked the revised portions in Red in the manuscript. We hope that the revised version of the manuscript is now acceptable for publication in Land. The main comments and our specific responses are attached below.
